# Anti-Cancer Effect of Chlorophyllin-Assisted Photodynamic Therapy to Induce Apoptosis through Oxidative Stress on Human Cervical Cancer

**DOI:** 10.3390/ijms241411565

**Published:** 2023-07-17

**Authors:** Seong-Yeong Heo, Yeachan Lee, Tae-Hee Kim, Soo-Jin Heo, Hwarang Shin, Jiho Lee, Myunggi Yi, Hyun Wook Kang, Won-Kyo Jung

**Affiliations:** 1Jeju Bio Research Center, Korea Institute of Ocean Science and Technology (KIOST), Jeju 63349, Republic of Korea; syheo@kiost.ac.kr (S.-Y.H.); sjheo@kiost.ac.kr (S.-J.H.); 2Marine Integrated Biomedical Technology Center, The National Key Research Institutes in Universities, Pukyong National University, Busan 48513, Republic of Korea; yclee@pukyong.ac.kr (Y.L.); taehee94@pknu.ac.kr (T.-H.K.); hwarangs@pukyong.ac.kr (H.S.); leejiho@pukyong.ac.kr (J.L.); 3Research Center for Marine Integrated Bionics Technology, Pukyong National University, Busan 48513, Republic of Korea; myunggi@pknu.ac.kr; 4Major of Biomedical Engineering, Division of Smart Healthcare and New-Senior Healthcare Innovation Center (BK21 Plus), Pukyong National University, Busan 48513, Republic of Korea

**Keywords:** chlorophyllin, photodynamic therapy, reactive oxygen species, apoptosis

## Abstract

Photodynamic therapy is an alternative approach to treating tumors that utilizes photochemical reactions between a photosensitizer and laser irradiation for the generation of reactive oxygen species. Currently, natural photosensitive compounds are being promised to replace synthetic photosensitizers used in photodynamic therapy because of their low toxicity, lesser side effects, and high solubility in water. Therefore, the present study investigated the anti-cancer efficacy of chlorophyllin-assisted photodynamic therapy on human cervical cancer by inducing apoptotic response through oxidative stress. The chlorophyllin-assisted photodynamic therapy significantly induced cytotoxicity, and the optimal conditions were determined based on the results, including laser irradiation time, laser power density, and chlorophyllin concentration. In addition, reactive oxygen species generation and Annexin V expression level were detected on the photodynamic reaction-treated HeLa cells under the optimized conditions to evaluate apoptosis using a fluorescence microscope. In the Western blotting analysis, the photodynamic therapy group showed the increased protein expression level of the cleaved caspase 8, caspase 9, Bax, and cytochrome C, and the suppressed protein expression level of Bcl-2, pro-caspase 8, and pro-caspase 9. Moreover, the proposed photodynamic therapy downregulated the phosphorylation of AKT1 in the HeLa cells. Therefore, our results suggest that the chlorophyllin-assisted photodynamic therapy has potential as an antitumor therapy for cervical cancer.

## 1. Introduction

Concurrent chemoradiation and external beam radiation are currently being tried for cervical cancer therapy [1]. However, radiation therapy yields an objective response less than 36% and significant toxicity after re-irradiation [2,3,4]. Photodynamic therapy (PDT) is a promising treatment method for cancers that do not respond to standard therapies [5,6]. Generally, photodynamic reaction induces tumor death by generating reactive oxygen species (ROS), including superoxide anion (O_2_^−^) and/or singlet oxygen (^1^O_2_), through photoactivation of photosensitizers [7,8]. It is a precisely controllable therapy and a minimally invasive or non-invasive combination of treatments that has garnered much attention in tumor treatment [9]. For these reasons, PDT has become the main subject of intense investigation due to a clinically promising approach to the treatment of cancer. In accordance with previous studies, PDT has been applied to treat neoplastic diseases, such as skin cancer [10], thyroid cancer [11], lung cancer [12], and breast cancer [13]. Moreover, PDT has been reported as an alternative approach for treating anti-viral, anti-bacterial, and antifungal activities for drug-resistant organisms [14,15]. Despite several benefits of PDT, current photosensitizers have limitations, such as aggregation in water, toxicity at high therapeutic concentrations, and long elimination half-life [16]. Thus, considerable attention has been paid to developing higher-efficacy photosensitizers with low toxicity from natural organisms.

Natural products have potential medicinal properties with lesser side effects compared with synthetic agents. Although many natural products are established for biological activities, they have not yet been investigated for photoactive properties [17]. Chlorophyllin is a semi-synthetic green plant pigment chlorophyll-derivative approved as a natural food additive and colorant in dietary supplements and cosmetics [18]. It has been a medicine for over 50 years without any adverse effects [19]. According to previous studies, chlorophyllin is easily soluble in water, has low toxicity, and is low cost compared to synthetic photosensitizers [20]. In particular, chlorophyllin has higher stability towards light, can locate in mitochondria, and is quickly cleared from the body [21]. Therefore, the purpose of the present study is to investigate the feasibility of chlorophyllin as a photosensitizer and to validate the anti-cancer effect of PDT with chlorophyllin on human cervical cancer. In this study, we demonstrated the potential therapeutic effects of the chlorophyllin-assisted PDT by evaluating ROS generation and apoptotic response in human cervical carcinoma HeLa cells.

## 2. Results

### 2.1. Cytotoxicity and Photo-Stimulated Cytotoxicity of Chlorophyllin in HeLa Cells

Prior to the phototoxic evaluation of chlorophyllin, we investigated its cytotoxic effect on macrophages and HeLa cells using MTT assay. As shown in Figure 1A, no significant cytotoxicity was observed in RAW 264.7 macrophages at concentrations of 0.05 to 4 μg/mL, but the high concentrations of chlorophyllin showed <90% cell viability. In addition, the cell viability of chlorophyllin in HeLa cells was determined at 0.05 to 4 μg/mL. In Figure 1B, chlorophyllin exhibited no cytotoxicity in HeLa cells.

Subsequently, the photo-stimulated cytotoxicity was evaluated using MTT assay. As shown in Figure 2A, approximately 90% of strong cytotoxicity was observed in chlorophyllin (2 and 4 μg/mL) with 800 mW/cm^2^ (48 J/cm^2^) of 405-nm laser after 24 h incubation. To determine the optimal conditions for the photodynamic reaction based on laser power densities, irradiation time, and chlorophyllin concentration, the cytotoxic effect was evaluated on HeLa cells using MTT assay. Most of the groups observed cytotoxicity of approximately 90%, whereas several of the evaluated groups had <90% cytotoxic values (Figure 2B). Based on the results in Figure 3, we selected appropriate conditions (2 μg/mL chlorophyllin with 800 mW/cm^2^ for 120 s and 4 μg/mL of chlorophyllin with 800 mW/cm^2^ for 30 s and 60 s), and we determined optimal conditions by live/dead assay. Compared with the no-treatment group, the cells with only laser irradiation or chlorophyllin treatment did not show obvious cytotoxicity, demonstrating that the laser irradiation or chlorophyllin had little effect on cytotoxicity (Figure 3). In contrast, groups of chlorophyllin-assisted PDT were observed that induced complete cell death by photo-stimulated cytotoxicity, except the group with 4 μg/mL chlorophyllin combined with laser irradiation (800 mW/cm^2^ for 30 s). Among the groups with photodynamic reactions, 4 μg/mL chlorophyllin combined with 48 J/cm^2^ (800 mW/cm^2^ for 60 s) laser irradiation was determined as the optimal condition, and hence was selected for the subsequent studies.

### 2.2. Measurement of Oxidative Stress after Laser Irradiation in Presence of Chlorophyllin

To evaluate the mechanism of cell death under photodynamic reaction, ROS generation was investigated when cells were stimulated with a laser in the absence or presence of chlorophyllin. The result showed that cells with chlorophyllin-assisted PDT markedly induced ROS generation, whereas there was decreased ROS generation from 40 min after laser irradiation (Figure 4). Moreover, the photodynamic reaction affected cell morphology by inducing cell shrinkage and membrane blebbing formation (Appendix A). In contrast, laser irradiation without chlorophyllin did not show any ROS generation or morphology change. In addition, we determined whether heat generated by chlorophyllin-assisted PDT affects the cells. As shown in the result, heat generated by laser irradiation was not efficient for inducing apoptosis (Figure 5).

### 2.3. Chlorophyllin-Assisted PDT Induced Apoptotic Cell Death on HeLa Cells

In order to determine the induction of apoptosis in the HeLa cells with photodynamic reaction, cell membrane phosphatidylserine content was evaluated by monitoring Annexin V/PI double staining. As shown in Figure 6, chlorophyllin-assisted PDT alone increased fluorescence intensities of annexin V and PI compared to other groups. Western blot in Figure 7 showed that chlorophyllin-assisted PDT downregulates protein expression levels of B-cell lymphoma-2 (BCL-2), which is an anti-apoptotic protein. On the other hand, chlorophyllin-assisted PDT upregulated protein expression levels of BCL-2 associated X protein (Bax) and cytochrome C, which are apoptotic-related proteins. Subsequently, we determined whether photodynamic reaction modulates the protein expression of caspase cascades. The results show that chlorophyllin-assisted PDT increased the cleaved caspase 8 and caspase 9, which are apoptosis-associated proteins. Therefore, these results indicate that chlorophyllin-assisted PDT induces caspase-dependent programmed cancer death.

### 2.4. Chlorophyllin-Assisted PDT Downregulates the AKT-1 Pathway

To elucidate the mechanism of the programmed cell death, we investigated the effect of photodynamic reaction on phosphorylation of AKT-1 expression in HeLa cells. Figure 8 shows that chlorophyllin-assisted PDT alone markedly downregulated the phosphorylation of AKT-1 compared with the other groups. Based on the results, it can be concluded that photodynamic reaction-induced apoptosis was associated with the regulation of AKT-1 pathways.

## 3. Discussion

PDT is the application of a laser using specific wavelengths for therapeutic responses, and it has been widely used for various clinical therapies [22,23]. It is especially applied to treat various cancers by eliciting photochemical and photophysical responses [24]. After the treatment of light sources with photosensitive agents, the release of ROS induced by photodynamic reaction lead to cell apoptosis or necrosis [25]. However, current photosensitizers are hydrophobic and toxic, and they have a long half-life time in the body [16]. Moreover, some defects have been reported for conventional photosensitizers, including phototoxic damage to adjacent tissues or ROS production insufficient to induce cell death [26,27]. Therefore, there is a crucial need to develop non-toxic, water-soluble, and short half-life agents for alternative photosensitizers that have minimal adverse effects.

Chlorophyllin, as a molecular analog of chlorophyll, is a semi-synthetic mixture of sodium copper complex [28]. It is hydrophilic in that the phytol tail is removed by the substitution of metal ion at the center of the porphyrin ring, whereas chlorophyll is hydrophobic [29]. It is quite advantageous to apply it in in vivo studies and show its clinical utility. Moreover, chlorophyllin is non-toxic, is quickly cleared from the body, and is more stable to the light and acid compared to chlorophyll [21,30]. As numerous studies have shown, chlorophyllin has several biological activities, including antioxidant [31], anti-proliferation [32], antibacterial [33], and anti-inflammatory properties [34]. In particular, the chlorophyllin has the main absorption peak at 405 nm, and can thus be applied as a photosensitizer (Appendix A). Thus, applications of a photodynamic reaction with chlorophyllin and a 405-nm laser to induce cell death were investigated.

In this study, we verified the anti-cancer effect of chlorophyllin combined with 405-nm laser irradiation on HeLa cells. According to the results, the maximal permissible dose of chlorophyllin was determined to consider the cytotoxicity of normal cells for overcoming the limitations of the previous photosensitizers. As shown in Figure 3, no cytotoxic effects were observed in HeLa cells treated with chlorophyllin alone, but strong cytotoxicity was observed after laser irradiation. These results indicate that 4 μg/mL chlorophyllin is effective in inducing the cell death only with photodynamic reaction. Moreover, intracellular ROS was observed in chlorophyllin-assisted PDT, implying that PDT-induced ROS is concerned with cell apoptosis (Figure 4). The intracellular ROS, which is generated by the photodynamic reaction, damages proteins, nuclei, membranes, and organelles to induce programmed cell death [35]. Many studies have investigated anti-cancer research using intracellular ROS [36,37,38]. In particular, intracellular ROS formed by photodynamic reaction can directly damage mitochondria, which act as key regulators in apoptosis [39]. Therefore, this study investigated the chlorophyllin-assisted PDT mediated induction of apoptosis on HeLa cells via mitochondrial dysfunction. However, excessive ROS production can damage not only cancer but also normal tissues and biological substances [40]. On the other hand, the PDT approach can be used in a local area, and the amount of ROS generation can be controlled by the laser intensity and the dose of the photosensitizer [41]. Therefore, PDT is currently being investigated as a new biotechnology approach to overcome the issue of various side effects in the field of cancer treatment.

BCL-2, as an anti-apoptotic protein, plays a role against the release of cytochrome C [42]. In contrast, Bax, a pro-apoptotic protein, is capable of suppressing the ability of BCL-2 to block apoptosis [43]. Therefore, a decline in BCL-2/Bax ratio is considered cell apoptosis, which releases mitochondrial cytochrome C via the mitochondrial pathway [44]. Cytochrome C is then released from mitochondria, which form an apoptosome with apoptosis-protease activating factor-1 and pro-caspase 9 [45]. After the formation of the apoptosome, pro-caspase 9 is converted to cleaved-caspase 9, initiating the mitochondria-mediated apoptosis pathway [46]. AKT pathway is associated with survival pathway regulating cell cycle progression, differentiation, apoptosis, and migration [47]. In particular, AKT pathway inhibits the activation of caspase 9, which is a key pro-apoptotic signaling molecule and mitochondrial apoptosis pathway [48]. According to Figure 7 and Figure 8, the chlorophyllin-assisted PDT induced downregulation of BCL-2/Bax proportion, but it increased the expression level of cytochrome C. Expression levels of cleaved caspase 8 and caspase 9 also increased after photodynamic reaction. Furthermore, the proposed PDT markedly decreased the phosphorylation of AKT-1. Based on the results, the chlorophyllin-assisted PDT causes apoptosis that induces ROS-mediated mitochondrial dysfunction through downregulating AKT-1 phosphorylation. Further in vivo tests will be conducted in the future to verify the efficacy and the safety of the chlorophyllin-assisted PDT, and to determine the optimal conditions for clinical application.

## 4. Materials and Methods

### 4.1. Materials

Chlorophyllin was obtained from EPC Natural Products (EPC Natural Products Co., Ltd., Beijing, China), and 3-(4,5-dimethylthiazol-2-yl)-2,5-diphenyltetrazolium bromide) (MTT) was purchased from Sigma-Aldrich (St. Louis, MO, USA). Dulbecco’s minimum Eagle’s medium (DMEM), fetal bovine serum (FBS), trypsin (250 U/mg), penicillin/streptomycin, and other materials used in the cell culture experiment were purchased from GIBCO^TM^, Invitrogen Corporation (Carlsbad, CA, USA). The specific antibodies used for Western blot analysis were purchased from Santa Cruz Biotechnology (Santa Cruz, CA, USA) and Amersham Pharmacia Biosciences (Piscataway, NJ, USA). All other chemicals and solvents were analytical grade, and the water used in the experiments was deionized.

### 4.2. Laser Light Source

A 405-nm diode laser system (FC-W-405-5W, CNI, Changchun, China) was used for PDT in the continuous-wave mode. A multimode flat optical fiber (core diameter = 600 μm, TeCure, Inc., Busan, Republic of Korea) was positioned 5.4 cm above the target surface of the prepared samples to achieve a spot size of 1.8 cm^2^. The laser output from the optical fiber was monitored using a power meter (Nova II, Ophir, Jerusalem, Israel) and a power sensor (50(150)A-BB-26, Ophir, Jerusalem, Israel) to ensure consistent experimental conditions. The temperature of the media was measured using an infrared camera (FLIRA300, FLIR System, Stockholm, Sweden) during laser irradiation.

### 4.3. Cell Culture

HeLa cell was obtained from Korea Cell Line Bank (Seoul, Republic of Korea). The cells were cultured in DMEM supplemented with 10% fetal bovine serum, 1% penicillin (100 U/mL), and streptomycin (100 μg/mL) at 37 °C in a 5% CO_2_ humidified environment.

### 4.4. Cytotoxicity Assessment

Cells were suspended in complete medium and seeded into 96-well plates at a density of 5 × 10^4^ cells/well. After seeding, the cells were treated with various concentrations (0.05 to 16 μg/mL) of chlorophyllin to evaluate the maximum permissible dosage of chlorophyllin. MTT solution was added to each well and incubated for 3 h. The medium was removed, and formazan crystals were dissolved in 200 μL dimethylsulfoxide (DMSO). The absorbance was measured with a microplate reader (PowerWave XS2, BioTek Instruments, Inc., Winooski, VT, USA) at 550 nm.

### 4.5. PDT on Hela Cells

For chlorophyllin-assisted PDT, Hela cells (5 × 10^4^ cells/well) were seeded in 96-well plates for 24 h and pre-determined concentrations of chlorophyllin were added. After 4 h, the medium was removed and rinsed twice with PBS. Subsequently, each plate was irradiated using 405-nm laser at different powers and irradiation times in order to evaluate the optimal conditions for cell death. Based on the cytotoxicity results, 4 μg/mL chlorophyllin and 48 J/cm^2^ (800 mW/cm^2^ for 60 s) laser were selected as the optimal conditions of photodynamic reaction and were used for further experiments.

### 4.6. Live/Dead Staining Assay

The live/dead staining assay was performed using the fluorescein diacetate/propidium iodide (FDA/PI). To observe cell viability, HeLa cells (5 × 10^4^ cells/well) were treated with chlorophyllin for 4 h, and then rinsed by PBS. Subsequently, cells were irradiated using 405 nm laser at different conditions and stained with FDA and PI after 24 h post irradiation. The stained cells were observed and qualitatively analyzed under a fluorescence microscope (Axio Observer A1, Zeiss, Jena, Germany).

### 4.7. Measurement of Intracellular ROS

The intracellular ROS generation was measured at 10 min to 100 min after laser irradiation using dichloro-dihydro-fluorescein diacetate (DCFH-DA). HeLa cells were seeded into 24-well plates 1 × 10^5^ cells/well and incubated for 24 h. Cells were treated with 4 μg/mL chlorophyllin for 4 h, and then rinsed twice with PBS. After being irradiated using 405-nm laser at 48 J/cm^2^, cells were incubated with 40 μM DCFH-DA for 30 min. Fluorescence microscopy was used to detect the generation of ROS.

### 4.8. Annexin V-FITC/Propidium Iodide (PI) Labeling

Fluorescein isothiocyanate (FITC) Annexin V Apoptosis Detection Kit (BD Pharmingen, San Jose, CA USA) was used to detect and measure apoptosis as described previously by Nguyen et al. (2019) [49]. HeLa cells were treated with chlorophyllin and 405-nm laser, as with the previous method at optimized conditions. After 24 h, medium was aspirated and rinsed by PBS. Cells were then treated with binding buffer containing Annexin V-FITC and PI in the dark for 30 min at room temperature. Each plate was rinsed twice with cold PBS and then observed with fluorescence microscopy.

### 4.9. Western Blot Analysis

The cells were lysed in lysis buffer (20 mM Tris, 5 mM EDTA, 10 mM Na_4_P_2_O_7_, 100 mM NaF, 2 mM Na_3_VO_4_, 1% NP-40, 10 mg/mL aprotinin, 10 mg/mL leupeptin, and 1 mM PMSF) for 60 min and then centrifuged at 12,000 rpm and 4 °C for 15 min. The protein concentrations were determined using the BCA^TM^ protein assay kit. The lysate containing 40 μg protein was subjected to electrophoresis on a sodium dodecyl sulfate (SDS)-polyacrylamide gel. The gel was then transferred onto a nitrocellulose membrane. The membrane was blocked with 5% nonfat dry milk in TBS-T (25 mM Tris-HCl, 137 mM NaCl, 2.65 mM KCl, and 0.05% Tween 20, pH 7.4) for 2 h. Primary antibodies were used at a 1:1000 dilution. The membranes were incubated with the primary antibodies at 4 °C overnight, washed with TBS-T, and then incubated with the secondary antibodies at 1:3000 dilutions. The signals were developed using an ECL Western blotting detection kit and quantified using Davinci K ChemiDoc imaging system (Young Hwa scientific Co., Ltd., Seoul, Republic of Korea). Image J software (software version 1.46; Wayne Rasband, NIH, Bethesda, MD, USA) was used to quantify protein expression.

### 4.10. Statistical Analysis

All quantitative data are presented as means ± standard deviation for at least three individual experiments conducted with fresh reagents. Statistical comparisons of the mean values were performed using analysis of variance (ANOVA) followed by Duncan’s multiple range test using SPSS Statistics 12.0 software (SPSS, Inc., Chicago, IL, USA). The differences in mean values were considered statistically significant for * *p* < 0.05 and ** *p* < 0.01.

## Figures and Tables

**Figure 1 ijms-24-11565-f001:**
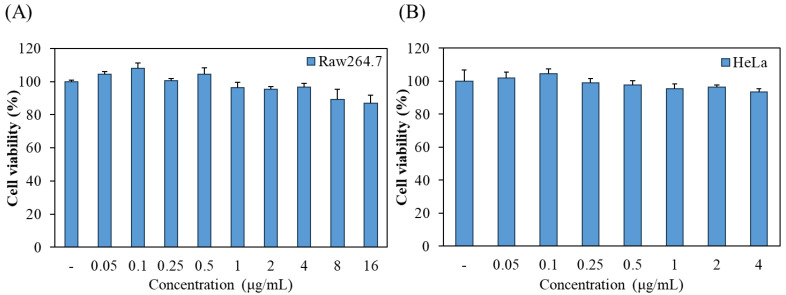
The effect of chlorophyllin on the cell viability of (**A**) Raw264.7 macrophages and (**B**) HeLa human cervical carcinoma cells. Cell viability was measured by MTT assay.

**Figure 2 ijms-24-11565-f002:**
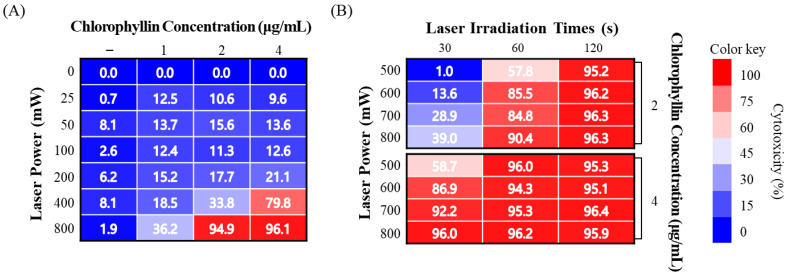
Phototoxicity of chlorophyllin with laser irradiation on HeLa cell. (**A**) Evaluation of cytotoxicity on various chlorophyllin concentrations and laser intensities for 60 s; (**B**) determination of the optimal condition of PDT on cell death.

**Figure 3 ijms-24-11565-f003:**
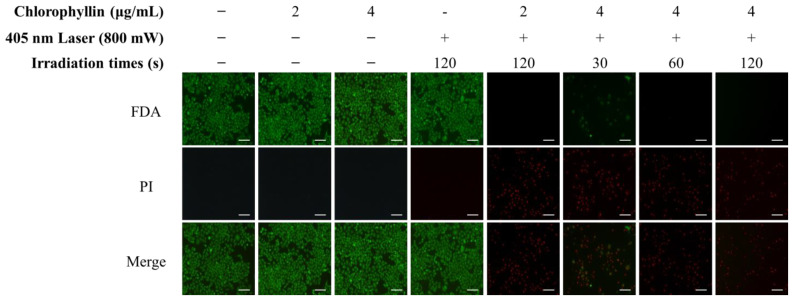
Phototoxicity of chlorophyllin-assisted PDT (405 nm, 800 mW/cm^2^) by live/dead assay. (green, live cells; red, dead cells). Scale bar, 200 μm.

**Figure 4 ijms-24-11565-f004:**
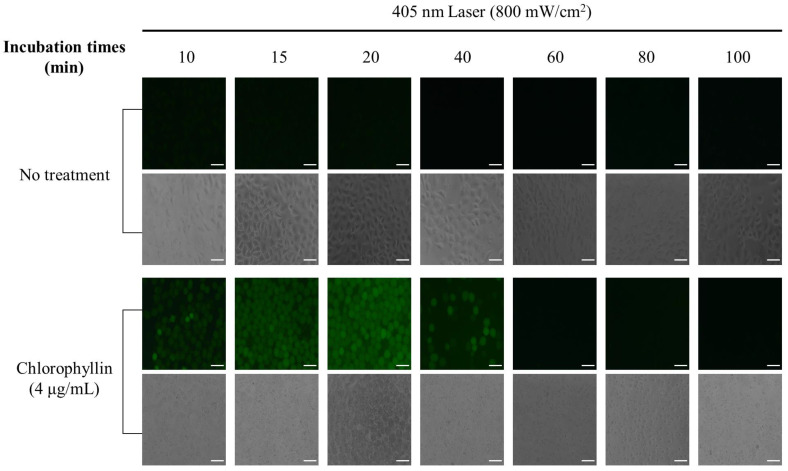
Chlorophyllin-assisted PDT in HeLa human cervical cells. ROS produced by irradiation of 405 nm laser (48 J/cm^2^) for 60 s. DCFH-DA (green) is used to label ROS production. Scale bar, 100 μm.

**Figure 5 ijms-24-11565-f005:**
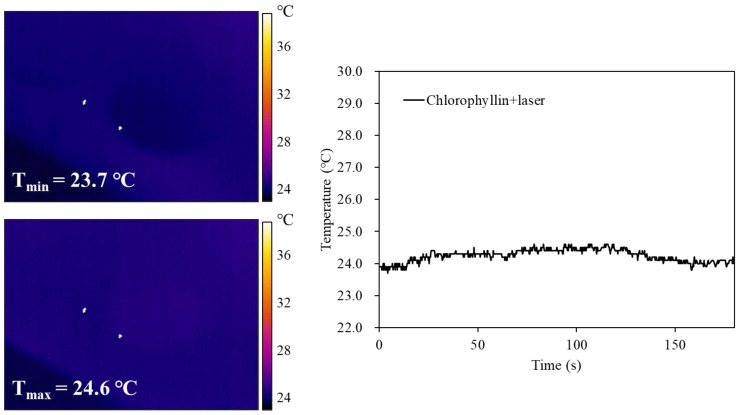
Thermal images and corresponding photothermal heating curves of chlorophyllin under continuous laser irradiation (405 nm, 800 mW/cm^2^) for 60 s, followed by natural cooling.

**Figure 6 ijms-24-11565-f006:**
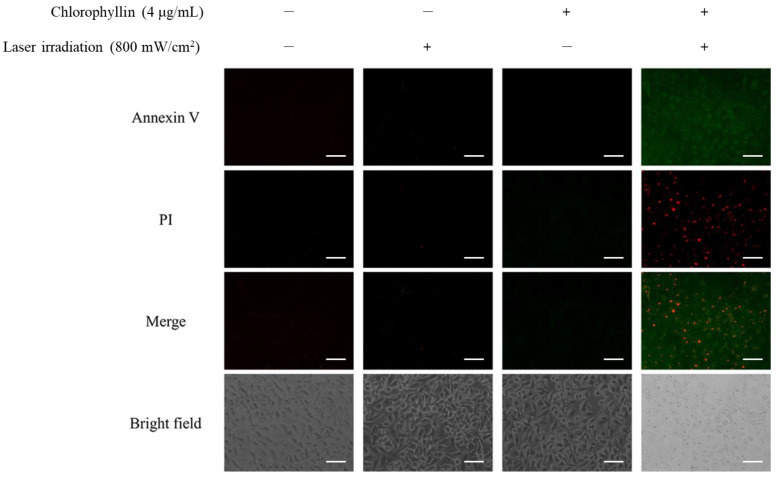
Chlorophyllin-assisted PDT on apoptosis with Annexin V-FITC detection kit by fluorescence microscopy. Annexin V/PI staining revealed that these cells were at an apoptosis stage (green and red fluorescence indicated Annexin V and PI, respectively). Scale bar, 100 μm.

**Figure 7 ijms-24-11565-f007:**
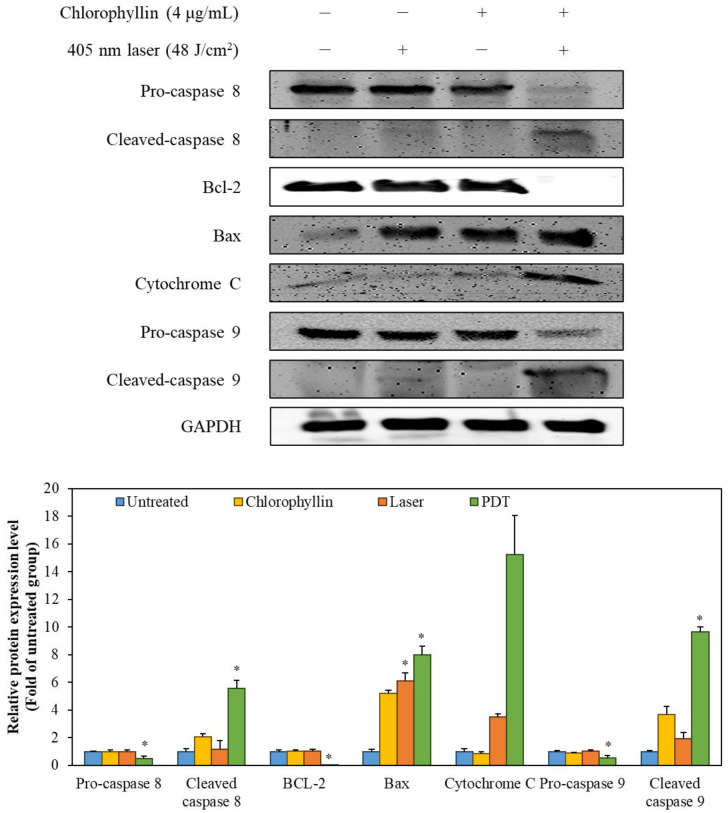
The expression of the apoptosis-related proteins was detected by Western blot analysis. The values are expressed as the means ± standard deviation of triplicate experiments. * *p* < 0.05 indicates significant differences compared with the untreated group.

**Figure 8 ijms-24-11565-f008:**
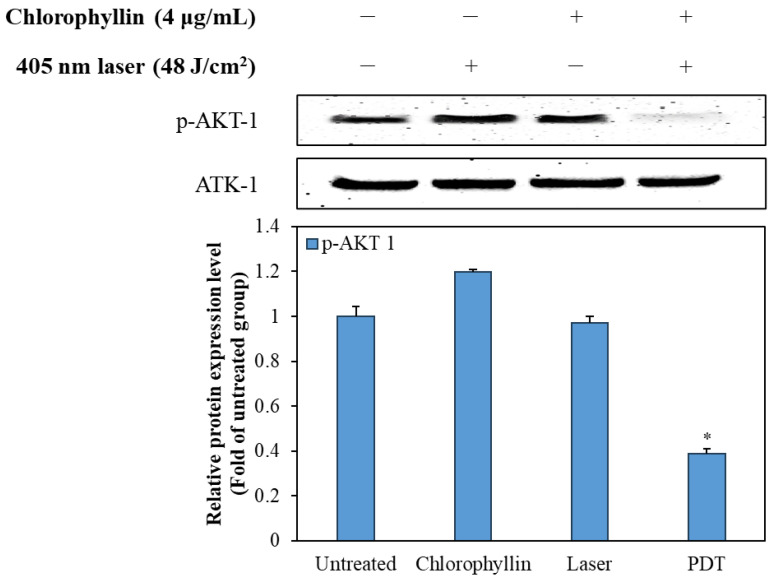
Effect of the chlorophyllin-assisted PDT on AKT-1 pathway in HeLa cells. The values are expressed as the means ± standard deviation of triplicate experiments. * *p* < 0.05 indicates significant differences compared with the untreated group.

## Data Availability

Data is contained within the article.

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
