# Peer review of "Anti-Cancer Effect of Chlorophyllin-Assisted Photodynamic Therapy to Induce Apoptosis through Oxidative Stress on Human Cervical Cancer"

_ijms, 2023, doi:10.3390/ijms241411565_

Round 1

Reviewer 1 Report

RAW 264.7 is a mouse cell and Hela is a human cell. Please show why you compared toxicity in these cells.

Also, although the concentrations being compared are different, please show why the concentrations shown in Figure 1 are sufficient to ensure safety.

Many studies on chlorophyllin and PDT have been reported.

For example:

/Photodiagnosis Photodyn Ther. 2017 Jun;18:162-170. doi: 10.1016/j.pdpdt.2017.01.186. epub 2017 Feb 24.

/Oncol Rep. 2019 Apr;41(4):2181-2193. doi: 10.3892/or.2019.7013. epub 2019 Feb 14.

Please clarify what is new about the authors' research compared to other studies.

Author Response

Response to Reviewer 1 Comments

Point 1: RAW 264.7 is a mouse cell and Hela is a human cell. Please show why you compared toxicity in these cells.

Also, although the concentrations being compared are different, please show why the concentrations shown in Figure 1 are sufficient to ensure safety.

Response 1:

Thank you for your comment. Previous studies have widely investigated the cytotoxic effect using a combination of mouse macrophage RAW264.7 and human cervical cancer Hela cell1,2, as well as human-derived cancer cells3,4. In addition, RAW264.7 is widely used in PDT research and is also used in cytotoxic evaluation5,6. Li et al. (2020) mentioned that improvements in the tumor specificity of photosensitizers, achieved through targeting or localized activation, could provide better outcomes with fewer adverse effects, as could combinations with chemotherapies or immunotherapies7. Therefore, the cytotoxic effect was evaluated to compare with RAW264.7 macrophage in this study.

Also as we mentioned in line 47, no cytotoxicity was observed at concentrations of 0.05 to 4 μg/ml, but 8 and 16 μg/ml of chlorophyllin showed below 90 % cell viability (Fig. 1A). According to Borawska et al. (2008)8, they considered that above 90% cell viability was no cytotoxicity. Therefore, we selected 4 μg/ml as the maximum concentration and proceeded with further experiments.

1 Lin, Yuan, et al. "Anti-inflammatory phenanthrene derivatives from stems of Dendrobium denneanum." Phytochemistry 95 (2013): 242-251.

2 Kanamori, Takashi, et al. "Coating lanthanide nanoparticles with carbohydrate ligands elicits affinity for HeLa and RAW264. 7 cells, enhancing their photodamaging effect." Bioorganic & Medicinal Chemistry 25.2 (2017): 743-749.

3 Gao, Hongwei, et al. "Tanshinones and diethyl blechnics with anti-inflammatory and anti-cancer activities from Salvia miltiorrhiza Bunge (Danshen)." Scientific Reports 6.1 (2016): 33720.

4 Song, Jihyeon, et al. "Formulation of glycyrrhizic acid-based nanocomplexes for enhanced anti-cancer and anti-inflammatory effects of curcumin." Biotechnology and Bioprocess Engineering 27.2 (2022): 163-170.

5 Kimura, Yuka, et al. "Photodynamic therapy using mannose-conjugated chlorin e6 increases cell surface calreticulin in cancer cells and promotes macrophage phagocytosis." Medical Oncology 39.6 (2022): 82.

6 Deng, Xiangyu, et al. "Polarization and function of tumor-associated macrophages mediate graphene oxide-induced photothermal cancer therapy." Journal of Photochemistry and Photobiology B: Biology 208 (2020): 111913.

7 Li, Xingshu, et al. "Clinical development and potential of photothermal and photodynamic therapies for cancer." Nature reviews Clinical oncology 17.11 (2020): 657-674.

8 Borawska, Maria H., et al. "Antimicrobial activity and cytotoxicity of picolinic acid and selected picolinates as new potential food preservatives." Polish journal of food and nutrition sciences 58.4 (2008).

Point 2: Many studies on chlorophyllin and PDT have been reported.

For example:

/Photodiagnosis Photodyn Ther. 2017 Jun;18:162-170. doi: 10.1016/j.pdpdt.2017.01.186. epub 2017 Feb 24.

/Oncol Rep. 2019 Apr;41(4):2181-2193. doi: 10.3892/or.2019.7013. epub 2019 Feb 14.

Please clarify what is new about the authors' research compared to other studies.

Response 2:

Thank you for your comment. In previous PDT studies using chlorophyllin, cytotoxicity, and apoptosis-related protein expression levels were analyzed, but not investigated apoptotic signaling pathways. In this study, the laser power and irradiation time that induce apoptosis were investigated and the optimal concentration of chlorophyllin was determined for apoptosis. Also, ROS and heat generation were investigated by the optimal conditions for chlorophyllin-PDT. In addition, we investigated the expression of apoptotic protein including cascade pathways and mechanisms of programmed cell death by chlorophyllin-assisted PDT.

Reviewer 2 Report

Reviewer comments and suggestions

The present study investigated the anti-cancer efficacy of chlorophyllin-assisted photodynamic therapy (PDT) on human cervical cancer by inducing apoptotic response through oxidative stress. The chlorophyllin-assisted PDT significantly induced cytotoxicity, and the optimal conditions were determined based on the results including laser irradiation time, laser power density, and chlorophyllin concentration. In the western blotting analysis, the PDT group displayed the increased protein expression level of the cleaved caspase 8, -9, Bax, and cytochrome C and the suppressed protein expression level of Bcl-2, pro-caspase 8, and -9. Additionally, the study reported that PDT downregulated the phosphorylation of AKT1 in the Hela cells. Hence, the result highlighted the chlorophyllin-assisted PDT has the potential as antitumor therapy for cervical cancer

Overall, the manuscript was well written. However, a few concerns/comments needed to be explained/modified. 

  1. Line 81-82 Is this correct?
  2. Comments for discussion: Please highlight the novelty of the study in the first paragraph rather than discussing n citing other studies
  3. Line 243 It should be uniform HeLa cells ( please check in the whole MS)
  4. All references should be modified based on the MDPI journal ( especially journal style format and year)

Author Response

Response to Reviewer 2 Comments

The present study investigated the anti-cancer efficacy of chlorophyllin-assisted photodynamic therapy (PDT) on human cervical cancer by inducing apoptotic response through oxidative stress. The chlorophyllin-assisted PDT significantly induced cytotoxicity, and the optimal conditions were determined based on the results including laser irradiation time, laser power density, and chlorophyllin concentration. In the western blotting analysis, the PDT group displayed the increased protein expression level of the cleaved caspase 8, -9, Bax, and cytochrome C and the suppressed protein expression level of Bcl-2, pro-caspase 8, and -9. Additionally, the study reported that PDT downregulated the phosphorylation of AKT1 in the Hela cells. Hence, the result highlighted the chlorophyllin-assisted PDT has the potential as antitumor therapy for cervical cancer

Overall, the manuscript was well written. However, a few concerns/comments needed to be explained/modified.

Point 1: Line 81-82 Is this correct?

Comments for discussion: Please highlight the novelty of the study in the first paragraph rather than discussing n citing other studies

Response 1:

In line 81-82, we have written the explanation for the results in Fig. 1. We described the contents of the reviewer’s comment, highlighting the novelty of the study, in the discussion section (line 50-65).

Line 50-65

Despite several benefits of PDT, current photosensitizers have limitations such as aggregation in water, toxicity at high therapeutic concentrations, and long elimination half-life. Thus, considerable attention has been paid to developing higher-efficacy photosensitizers with low toxicity from natural organisms.

Natural products have potential medicinal properties with lesser side effects compared with synthetic agents. Although many natural products are established for biological activities, they have not been yet investigated for photoactive properties. Chlorophyllin is a semi-synthetic green plant pigment chlorophyll-derivative approved as a natural food additive and colorant in dietary supplements and cosmetics. It has been a medicine for over 50 years without any adverse effects. According to previous studies, chlorophyllin is easily soluble in water, has low toxicity, and is low cost compared to synthetic photosensitizers. Especially, chlorophyllin has higher stability towards light, can locate in mitochondria, and is quickly cleared from the body. Therefore, the purpose of the present study is to investigate the feasibility of chlorophyllin as a photosensitizer and validate the anti-cancer effect of PDT with chlorophyllin on human cervical cancer.

Point 2: Line 243 It should be uniform HeLa cells ( please check in the whole MS)

Response 2:

According to the reviewer’s comment, we have changed ‘hela’ to ‘Hela’.

Point 3: All references should be modified based on the MDPI journal ( especially journal style format and year)

Response 2:

According to the reviewer’s comment, we have changed the reference form.

Reviewer 3 Report

This work is devoted to the current topic of creating biocompatible photosensitizers for photodynamic therapy of cancer. The authors used a commercial drug chlorophyllin and obtained data on its cytotoxicity and ability to induce oxidative stress under laser irradiation. Despite the relevance and importance of this work, there are a number of serious comments:

1) It is necessary to check the test of the article by an English-speaking native speaker. The text contains a number of typos and disagreements, or even missing verbs.

2) Figure 2. The authors should add to the caption of the figure the indication of sub-pictures A and B. The authors should indicate against which cells cytotoxicity was studied.

3) Page 2. Lines 80-85. The authors discuss the expressed. Cytotoxicity of chlorophyllin under the influence of laser radiation and in doing so refer to Figure 1.

4) Figure 3: The authors should specify what they mean by the term "cytotoxicity" (Percentage of surviving cells or percentage of dead cells?).

5) Also, in the caption of Figure 3, the authors should add a caption to the sub-figures A and B.

6) It is not clear to which laser exposure time the data in Figure 2-A refer and to which chlorophyll concentrations the data obtained by varying the exposure time in Figure 2-B refer. The authors should substantially modify this figure.

7) General comment to all figures: the font is not uniform, very small. The design of the figures is of poor quality and does not correspond to the high quartile of the IJMS journal.

8) The authors should also necessarily supplement the data presented in Figure 3 with a diagram reflecting the quantitative characteristics of visualized cells with mandatory statistical treatment. Visually, under conditions of 4 µg/ml, 405 nm, 800mW, 30 sec, the dead cells are much more detected than at longer times of exposure to the laser. However, the authors do not explain it.

9) It is required to correctly and uniformly label the HeLa cell line in the text.

10) There are a lot of abbreviations in the text, which the authors do not decipher. 

11) Figure 4. The authors should indicate the time of laser exposure to the cells. The indicated cell incubation time does not correspond to the time of direct exposure of cells to the laser.  

12) The first half of the section "discussion of results" repeats and expands the information presented in the introduction. The authors should provide in this section a detailed description of the discussion of the obtained results with their correlation with the existing ideas about the nature of the observed processes and phenomena.

13) Reference 34 in the text is a misprint.

14) Section 4.1. The authors say that they used the dye Hoechst 3342, although there is no data on its use in the text and figures.

15) The chlorophyllin used by the authors is practically not characterized (except for the absorption spectrum measured under unknown conditions). Characterization (absorption spectrum (in water or phosphate buffer), elemental analysis, IR spectroscopy) of chlorophyllin should be given in the text of the article.

It is necessary to have the test article checked by a native English speaker. There are a number of mistakes and inconsistencies in the text, or even missing verbs.

Author Response

Response to Reviewer 3 Comments

This work is devoted to the current topic of creating biocompatible photosensitizers for photodynamic therapy of cancer. The authors used a commercial drug chlorophyllin and obtained data on its cytotoxicity and ability to induce oxidative stress under laser irradiation. Despite the relevance and importance of this work, there are a number of serious comments:

Point 1: It is necessary to check the test of the article by an English-speaking native speaker. The text contains a number of typos and disagreements, or even missing verbs.

Response 1:

Thank you for your comment. We have already completed the English editing process and submitted the certification. In addition, we reviewed and revised the manuscript.

Point 2: Figure 2. The authors should add to the caption of the figure the indication of sub-pictures A and B. The authors should indicate against which cells cytotoxicity was studied.

Response 2:

It was a mistake while preparing the manuscript. According to the reviewer’s comment, we have modified the figure caption as below.

Line 99; Figure 2. Phototoxicity of chlorophyllin with laser irradiation on Hela cell. (A) evaluation of cytotoxicity on various chlorophyllin concentrations and laser intensities for 60 s, (B) Determination of the optimal condition of PDT on cell death

Point 3: Page 2. Lines 80-85. The authors discuss the expressed. Cytotoxicity of chlorophyllin under the influence of laser radiation and in doing so refer to Figure 1.

Response 3:

It was a mistake while preparing the manuscript. According to the reviewer’s comment, we have changed the sentence as below.

Line 81; As shown in Fig. 2A, approximately 90% of strong cytotoxicity was observed in chlorophyllin (2 and 4 μg/ml) with 800 mW/cm2 (48 J/cm2) of 405-nm laser after 24 h incubation.

Point 4: Figure 3: The authors should specify what they mean by the term "cytotoxicity" (Percentage of surviving cells or percentage of dead cells?).

Response 4:

Thank you for your comment. However, the groups of chlorophyllin-assisted PDT observed cells detachment from the culture plate because of strong cytotoxicity. Therefore, quantitative results of live/dead assay are difficult to provide in this study. Many previous studies also presented only fluorescence images without quantitative results. So, we described the cell state according to the fluorescence color in Figure 3 caption as below.

Line 104; Figure 3. Phototoxicity of chlorophyllin-assisted PDT (405 nm, 800 mW/cm2) by live/dead assay. (green, live cells; red, dead cells)

Point 5: Also, in the caption of Figure 3, the authors should add a caption to the sub-figures A and B.

Response 5:

Thank you for your comment. Figure 3 is a single figure, not consisting of subfigures. Therefore, the caption of sub-figures is not needed. In addition, it is more advantageous to interpret and explain the result by expressing it as a single figure rather than making two sub-figures.

Point 6: It is not clear to which laser exposure time the data in Figure 2-A refer and to which chlorophyll concentrations the data obtained by varying the exposure time in Figure 2-B refer. The authors should substantially modify this figure.

Response 6:

Thank you for your comment. We have corrected mistyping ‘as shown in Fig. 1’ to ‘as shown in Fig. 2A’ in line 81. Also, we described the selected conditions of chlorophyllin concentration and laser power to induce cell apoptosis in optimal conditions as below. In addition, we mentioned ‘chlorophyllin concentration’ on the right side of the Fig. 2B.

Line 81; As shown in Fig. 2A, approximately 90% of strong cytotoxicity was observed in chlorophyllin (2 and 4 μg/ml) with 800 mW/cm2 (48 J/cm2) of 405-nm laser after 24 h incubation.

Point 7: General comment to all figures: the font is not uniform, very small. The design of the figures is of poor quality and does not correspond to the high quartile of the IJMS journal.

Response 7:

According to the reviewer’s comment, we have modified the figures.

Point 8: The authors should also necessarily supplement the data presented in Figure 3 with a diagram reflecting the quantitative characteristics of visualized cells with mandatory statistical treatment. Visually, under conditions of 4 µg/ml, 405 nm, 800mW, 30 sec, the dead cells are much more detected than at longer times of exposure to the laser. However, the authors do not explain it.

Response 8:

Thank you for your comment. However, the groups of chlorophyllin-assisted PDT observed cells detachment from the culture plate because of strong cytotoxicity. Therefore, quantitative results of live/dead assay are difficult to provide in this study. Many previous studies also presented only fluorescence images without quantitative results. Please consider. In addition, we described the reviewer's mention in line 92-94 as below. Briefly, all chloropyllin-PDT groups were showed complete cell death, but viable cells were observed in small amounts under conditions of 4 µg/ml, 405 nm, 800mW, 30 sec. Based on the results, we selected the optimal condition.

Line 92; In contrast, groups of chlorophyllin-assisted PDT were observed that induced complete cell death by photo-stimulated cytotoxicity, except group with 4 μg/ml chlorophyllin combined with laser irradiation (800 mW/cm2 for 30 s).

Point 9: It is required to correctly and uniformly label the HeLa cell line in the text.

Response 9:

Thank you for your comment. According to the reviewer’s comment, we have corrected it in the manuscript.

Point 10: There are a lot of abbreviations in the text, which the authors do not decipher.

Response 10:

According to the reviewer’s comment, we have provided a long form to explain the abbreviation as below.

Line 130; B-cell lymphoma-2 (BCL-2)

Line 131; BCL-2 associated X protein (Bax)

Line 266; dichloro-dihydro-fluorescein diacetate (DCFH-DA)

Line 272; Fluorescein isothiocyanate (FITC)

Point 11: Figure 4. The authors should indicate the time of laser exposure to the cells. The indicated cell incubation time does not correspond to the time of direct exposure of cells to the laser.

Response 11:

According to the reviewer’s comment, we have corrected Figure 4 caption as below.

Line 118; Figure 4. Chlorophyllin-assisted PDT in Hela human cervical cells. ROS produced by irradiation of 405 nm laser (48 J/cm2) for 60 s. DCFH-DA (green) is used to label ROS production.

Point 12: The first half of the section "discussion of results" repeats and expands the information presented in the introduction. The authors should provide in this section a detailed description of the discussion of the obtained results with their correlation with the existing ideas about the nature of the observed processes and phenomena.

Response 12:

Thank you for your comment. According to the reviewer’s comment, we have corrected discussion section as below.

Line 188: Many studies have investigated anti-cancer research using producing intracellular ROS [36-38]. Especially, intracellular ROS formed by photodynamic reaction can directly damage mitochondria, which act as key regulators in apoptosis [36]. Therefore, this study investigated that the chlorophyllin-assisted PDT mediated its induction of apoptosis on HeLa cells via mitochondrial dysfunction. However, excessive ROS production can damage not only cancer also normal tissues and biological substances [39]. On the other hand, the PDT approach can be used in a local area, and the amount of ROS generation can be controlled by the laser intensity and the dose of the photosensitizer [40]. Therefore, PDT is currently being investigated as a new biotechnology approach to overcome the issue of various side effects in the field of cancer treatment.

Point 13: Reference 34 in the text is a misprint.

Response 13:

It was a mistake while preparing the manuscript. According to the reviewer’s comment, we have corrected it.

Point 14: Section 4.1. The authors say that they used the dye Hoechst 3342, although there is no data on its use in the text and figures.

Response 14:

It was a mistake while preparing the manuscript. According to the reviewer’s comment, we have corrected it.

Point 15: The chlorophyllin used by the authors is practically not characterized (except for the absorption spectrum measured under unknown conditions). Characterization (absorption spectrum (in water or phosphate buffer), elemental analysis, IR spectroscopy) of chlorophyllin should be given in the text of the article.

Response 15:

Thank you for your comment. Chlorophyllin is a well-known ingredient, and a lot of physicochemical characterizations including elemental analysis, IR spectroscopy have been investigated in previous studies. In this study, we evaluated the anti-cancer effect and signaling pathway by the chlorophyllin-assisted PDT reaction. Chlorophyllin was purchased from EPC Natural Products Co. as mentioned in line 217. The company provides documents like MSDS and CoA for chlorophyllin and guarantees the product. Therefore in this study, we uploaded the absorption of chlorophyllin result in supplemental data because the absorption wavelength of chlorophyllin is important to select laser wavelength.

Round 2

Reviewer 3 Report

The authors corrected almost all of the comments on the paper, with the exception of the drawings. It is necessary to improve the figures, namely:

1) Authors should unify their color design (some of the drawings are presented in shades of gray, some are colored figures)

2) Authors should increase and unify the font on all figures.

Author Response

Response to Reviewer 3 Comments

The authors corrected almost all of the comments on the paper, with the exception of the drawings. It is necessary to improve the figures, namely:

Point 1:  Authors should unify their color design (some of the drawings are presented in shades of gray, some are colored figures)

Response 1:

Thank you for your comment. According to the reviewer's comment, we have changed that figure colors have been unified.

Point 2: Authors should increase and unify the font on all figures.

Response 2:

Thank you for your comment. According to the reviewer's comment, we have changed that all figures style and typeface have been unified and increased font size.
